# Maternal Hypermethylated Genes Contribute to Intrauterine Growth Retardation of Piglets in Rongchang Pigs

**DOI:** 10.3390/ijms25126462

**Published:** 2024-06-12

**Authors:** Pingxian Wu, Junge Wang, Xiang Ji, Jie Chai, Li Chen, Tinghuan Zhang, Xi Long, Zhi Tu, Siqing Chen, Lijuan Zhang, Ketian Wang, Liang Zhang, Zongyi Guo, Jinyong Wang

**Affiliations:** 1Chongqing Academy of Animal Sciences, Rongchang, Chongqing 402460, Chinachensiqingcsq@163.com (S.C.);; 2National Center of Technology Innovation for Pigs, Rongchang, Chongqing 402460, China; 3Chongqing Modern Agricultural Industry Technology System, Chongqing 401120, China; 4Farm Animal Genetic Resources Exploration and Innovation Key Laboratory of Sichuan Province, Sichuan Agricultural University, Chengdu 611130, China

**Keywords:** Rongchang pigs, intrauterine growth retardation, DNA methylation, transcriptome, placenta

## Abstract

The placenta is a crucial determinant of fetal survival, growth, and development. Deficiency in placental development directly causes intrauterine growth retardation (IUGR). IUGR can lead to fetal growth restriction and an increase in the mortality rate. The genetic mechanisms underlying IUGR development, however, remain unclear. In the present study, we integrated whole-genome DNA methylation and transcriptomic analyses to determine distinct gene expression patterns in various placental tissues to identify pivotal genes that are implicated with IUGR development. By performing RNA-sequencing analysis, 1487 differentially expressed genes (DEGs), with 737 upregulated and 750 downregulated genes, were identified in IUGR pigs (H_IUGR) compared with that in normal birth weight pigs (N_IUGR) (*p* < 0.05); furthermore, 77 miRNAs, 1331 lncRNAs, and 61 circRNAs were differentially expressed. The protein–protein interaction network analysis revealed that among these DEGs, the genes GNGT1, ANXA1, and CDC20 related to cellular developmental processes and blood vessel development were the key genes associated with the development of IUGR. A total of 495,870 differentially methylated regions were identified between the N_IUGR and H_IUGR groups, which included 25,053 differentially methylated genes (DMEs); moreover, the overall methylation level was higher in the H_IUGR group than in the N_IUGR group. Combined analysis showed an inverse correlation between methylation levels and gene expression. A total of 1375 genes involved in developmental processes, tissue development, and immune system regulation exhibited methylation differences in gene expression levels in the promoter regions and gene ontology regions. Five genes, namely, ANXA1, ADM, NRP2, SHH, and SMAD1, with high methylation levels were identified as potential contributors to IUGR development. These findings provide valuable insights that DNA methylation plays a crucial role in the epigenetic regulation of gene expression and mammalian development and that DNA-hypermethylated genes contribute to IUGR development in Rongchang pigs.

## 1. Introduction

Enhancing litter size and birth weight is a prominent research objective of the swine industry and swine breeders. Several factors influence litter size, including ovulation rate, fetal survival rate, and uterine capacity. Intrauterine growth retardation (IUGR) represents the adverse effects experienced by mammalian fetuses in the uterus, resulting in restricted fetal growth and development. This condition manifests as hypoplasia of functional organs, particularly the placenta, and is externally evident as a remarkable reduction in the body weight of normal-age fetuses [1]. As multiparous mammals, pigs are often affected by IUGR [2]. During the embryonic development of pigs, factors such as maternal malnutrition, health conditions, genetic influences, and environmental stress can contribute to IUGR. For instance, the addition of L-arginine to the diet of pregnant sows during the latter stages of gestation has been shown to enhance birth weight in piglets, thereby mitigating the occurrence of IUGR [3]. Hu and colleagues [4] confirmed that Nox2 is highly expressed in the placenta of low-birth-weight piglets. It reduces placental angiogenesis through a mechanism dependent on mitochondrial ROS-STAT3-VEGF-A. This may be one of the contributing factors to IUGR. Moreover, IUGR piglets have high morbidity and mortality rates, impaired growth, and suboptimal carcass quality throughout their growth and development stages [1]; this condition may emerge in an asymmetrical (70–80% of cases) or symmetrical (20–30% of cases) manner [5]. Offspring with IUGR exhibit long-term complications, including increased susceptibility to metabolic syndrome, cardiovascular diseases, and type 2 diabetes, in adulthood [6].

The placenta plays a crucial role as a pivotal intermediary between the mother and the fetus, as it enables the transfer of essential nutrients, oxygen, and bioactive substances from maternal circulation to fetal circulation and simultaneously facilitates the elimination of metabolic waste products from the fetus through maternal circulation [7]. As a parenchymal organ, the placenta exhibits endocrine function and can synthesize bioactive substances, including hormones and enzymes, which play a regulatory role in maternal physiological functions throughout pregnancy [8]. Placental dysfunction not only poses risks to maternal health but also serves as a critical contributing factor to fetal IUGR. As the fetus undergoes development throughout the entire gestational period, there is an elevated demand for nutrients and oxygen, thereby necessitating a substantial increase in blood flow within the placenta to meet fetal requirements [9]. Scientific studies have established a direct association between the birth weight of piglets and the density of the formed placental blood vessels. Specifically, a reduced density of placental blood vessels is associated with a decreased number and surface area of villi, resulting in lower birth weight in piglets [10]. Placental aging (premature maturation) may result in placental insufficiency, failing to support normal fetal growth and development sufficiently [11]. Furthermore, several diseases can impair placental function, impacting fetal growth and development. Examples include chronic interstitial chorioamnionitis, type 2 porcine reproductive and respiratory syndrome virus infection, and heat stress [12,13,14].

In mammals, DNA methylation occurs predominantly on the cytosine residues of CpG dinucleotides, and except for CpG islands (CpGIs) and other gene regulatory sequences, approximately 70–80% of all CpG sites are methylated [15]. Dynamic alterations in DNA methylation play a critical role in the epigenetic regulation of gene expression. Specifically, DNA methylation at gene promoters is known for its involvement in gene silencing, which is achieved by blocking the binding of transcription factors [16]. Methylation at different positions in the gene structure may have different regulatory effects on gene expression. Intergenic methylation may affect gene expression and transcription activity by regulating the binding of distal enhancers [17]. However, scarce research has been conducted to investigate genome-wide methylation at the single-nucleotide resolution.

Pigs are a major contributor to meat production and have emerged as an exceptional animal model to investigate human development and disease [18]. Therefore, in the present study, we conducted gene-wide transcriptomics and genome-wide methylation analyses on placental tissues from normal piglets and IUGR piglets. We also performed a correlation analysis using differentially methylated regions (DMRs) and mRNA pairs and observed a notable negative correlation between methylation changes in genes and gene expression levels. The investigation of porcine IUGR is a critical topic as the findings can reveal the molecular mechanisms underlying the regulation of mammalian IUGR and could serve as a valuable resource for research on animal breeding, reproductive biology, and related diseases.

## 2. Results

### 2.1. Statistical Analysis

Table 1 summarizes the phenotypic statistics of piglets corresponding to each placental tissue. An IUGR piglet was defined as a piglet with a birth weight lower than two standard deviations from the mean birth weight of all piglets [1,19]. IUGR piglets born to sows with a history of IUGR were assigned to the H_IUGR group, and normal piglets born to sows without previous IUGR episodes were assigned to the N_IUGR group. The average body weights of the N_IUGR and H_IUGR groups were 0.77 ± 0.03 and 0.45 ± 0.02 kg, respectively. Importantly, the body weight of all IUGR piglets was within two standard deviations of the mean weight of normal-weight piglets; this finding was in line with the established guidelines [1,19].

### 2.2. Comparative Analysis of the Characteristics of lncRNAs and mRNAs

A total of 927,907,980 raw reads were generated from the cDNA sequencing libraries prepared for 10 placental tissues. Following quality control, 906,758,804 high-quality reads were retained, and the proportion of high-quality reads to raw reads ranged from 97.32% to 98.21% across the 10 samples. To further investigate the characteristic differences between lncRNAs and miRNAs in the 10 placental tissues, a comparative analysis was conducted on the transcript length, number of exons, open reading frame (ORF) length, and expression levels of the identified lncRNAs and miRNAs. As shown in Figure 1, the average length of lncRNAs (3324 nt) was significantly shorter than that of mRNAs (3513 nt), and the average number of exons for lncRNAs (2.83) was substantially lower than that for mRNAs (12.27). The analysis of the ORF length showed that the average ORF length of lncRNAs (108 nt) was considerably shorter than that of mRNAs (604 nt). Gene expression levels, reflected by gene abundance, indicated higher expression levels with greater gene abundance. The FPKM method was used to assess the expression levels of lncRNAs and mRNAs in the two groups; the results showed that mRNA expression levels were relatively higher than those of lncRNAs.

### 2.3. Detailed Functional Profiles of the DEGs between the H_IUGR and N_IUGR Groups

Functional analyses, including Gene Ontology (GO) and Kyoto Encyclopedia of Genes and Genomes (KEGG) pathway enrichment analysis, were conducted to determine the function of DEGs between the H_IUGR and N_IUGR groups to reveal potential physiological alterations in IUGR placental tissues. A comparative analysis was performed on the DEGs between the placental tissues of the two groups. The results revealed 1487 DEGs, which included 737 upregulated genes and 750 downregulated genes (Figure 2A). These DEGs exhibited significant enrichment in 17 GO terms (corrected *p* < 0.05), including 10 biological processes, five cellular components, and two molecular functions. The significantly enriched GO terms were primarily related to developmental processes, cell development processes, blood vessel development, and tissue development (Appendix A); these genes are known to be associated with organismal growth and development, thus suggesting their potential effect on IUGR piglets. The 1487 identified DEGs were then evaluated by a KEGG pathway enrichment analysis. Among the enriched pathways, tight junctions, metabolic pathways, and carbon metabolism were associated with the growth and development of piglets. Furthermore, 77 miRNAs (41 upregulated and 36 downregulated miRNAs), 1331 lncRNAs (893 upregulated and 438 downregulated lncRNAs), and 61 circRNAs (39 upregulated and 22 downregulated circRNAs) were differentially expressed (Figure 2B–D).

### 2.4. Protein–Protein Interaction Network of the DEGs

The Search Tool for the Retrieval of Interacting Genes (STRING) database (http://string-db.org/) is an online tool for assessing gene interactions and enables the analysis of protein–protein interaction (PPI) networks. The resulting network was visualized with Cytoscape (Version 3.5.1) software. The PPI network analysis revealed several hub genes, including GNGT1, ANXA1, and CDC20 (Figure 3). These genes are strongly associated with cellular proliferation, cell division, and inflammatory response.

### 2.5. Genome-Wide DNA Methylation Profiling of Placental Tissues

We anticipated variations in DNA methylation levels between the two distinct placental tissues. Appendix A shows the quality of clean reads and the bisulfite conversion efficiency for all samples. The bisulfite conversion efficiency exceeded 99.7% for each sample group. The alignment of clean reads to the pig reference genome ranged from 62.50% to 76.50% of the total reads, thus indicating satisfactory alignment rates. The coverage of C sites in the sequenced data is a crucial indicator of methylation sequencing quality and depth. The examination of C sites in the three-sequence context revealed that genomic DNA cytosine in these 10 samples exhibited methylation in the range of 1.66–3.23%, with the highest proportion observed at CG sites. We analyzed the genome-wide distribution of DNA methylation and observed similar methylation patterns across the functional regions in both sample groups (Figure 4A). Moreover, the methylation level was higher in the H_IUGR group than in the N_IUGR group, thus suggesting a potential association between DNA methylation levels and IUGR occurrence. We then evaluated methylation levels in gene body regions and other functional elements. Gene bodies exhibited higher methylation levels than neighboring intergenic regions, while prominent hypomethylation was observed around transcription start sites (TSSs). Methylation levels in gene bodies remained relatively stable and showed a significant decrease after the transcription end site (TES). Promoters and CpGIs displayed low CpG methylation levels, whereas genes and intergenic regions exhibited higher methylation levels. We also conducted differential methylation analysis on placental samples and identified 495,870 DMRs. The length distribution of these DMRs ranged from 20 to 400 bp, with the majority of DMRs being less than 500 bp in length (Figure 4B). The alignment of DMRs to various genomic elements revealed that most DMRs were located in intron regions, followed by 5′-untranslated regions (5′-UTRs) and exon regions (Figure 4C).

### 2.6. Identification of Differentially Methylated Genes and Functional Enrichment Analysis

We compared the obtained 495,870 DMRs to the pig reference genome and found the involvement of 25,053 differentially methylated genes (DMGs). The main GO terms enriched in the DMGs between the H_IUGR and N_IUGR groups were identified (Figure 5A). The main GO items enriched included G protein-coupled receptor activity, olfactory receptor activity, and nucleosome, while the main KEGG pathways enriched included protein digestion and absorption, the PI3K-Akt signaling pathway, and Aminoacyl-tRNA biosynthesis (Figure 5B).

The promoter region plays a vital role in regulating gene expression. Several studies have demonstrated that the presence of DNA methylation, an epigenetic modification mechanism, in a gene’s promoter region has an inhibitory effect on its expression. To gain further insights into the impact of promoter region methylation on IUGR occurrence in piglets, this study conducted an enrichment analysis that specifically focused on genes exhibiting methylation in the promoter region. Among all DMRs, 11,198 genes were identified to be methylated in the promoter region. The enriched GO terms associated with these genes included structural constituents of the cytoskeleton, chromatin binding, regulation of transcription (DNA-templated), and transcription (DNA-templated). Notably, the majority of the enriched GO terms were closely associated with transcriptional regulation and exclusively occurred in the promoter region; this finding implied a potential relationship between methylation and gene regulation (Figure 5C). KEGG enrichment analysis was then conducted on the genes with differentially methylated promoter regions between the H_IUGR and N_IUGR groups. The results revealed significant enrichment in various pathways, primarily including the MAPK signaling pathway, and the ubiquitin-mediated proteolysis signaling pathway (Figure 5D).

### 2.7. Combined Analysis of Placental Transcriptome Sequencing and Genome-Wide Methylation Sequencing

We then performed an analysis of the correlation between DNA methylation and gene expression. A correlation analysis was performed to assess the relationship between DNA methylation levels in both the gene body and promoter regions, and the overall gene expression levels were normalized. Figure 6 shows the findings from two different placental tissues and two distinct methylated regions where high methylation levels were associated with lower gene expression levels.

We also assessed the methylation levels of upregulated and downregulated genes across various functional elements of the genome to determine the methylation extent in genes with different expression patterns. The expression levels of the upregulated genes and the methylation levels were higher in the H_IUGR group than in the N_IUGR group (Figure 7). Moreover, the methylation levels of these genes were lowest in the promoter region. We also analyzed the distribution densities of the upregulated and downregulated genes in the methylation sites. Interestingly, the methylation densities of the upregulated and downregulated genes in both groups were similar, thus suggesting that the methylation distribution density may not substantially affect gene expression.

To explore the potential functions of genes with differential methylation status between the H_IUGR and N_IUGR groups, enrichment analysis was conducted for genes with DMRs in the promoter regions and GO regions. A Venn diagram was generated to analyze the overlap between the obtained DMGs and DEGs based on the classification of high/low differential methylation and upregulated/downregulated gene expression. As depicted in Figure 8, 1377 genes exhibited simultaneous differences in methylation distribution and gene expression. These genes can be categorized into three types as follows: 402 genes that potentially engage in both positive and negative regulatory relationships (intersection of hyper-DMGs, hypo-DMGs, and up-DMGs, as well as the intersection of hyper-DMGs, hypo-DMGs, and down-DMGs); 384 genes that solely display positive regulatory relationships (intersection of hyper-DMGs with up-DMGs and intersection of hypo-DMGs with down-DMGs); and 591 genes that exclusively demonstrate negative regulatory relationships (intersection of hyper-DMGs with down-DMGs and intersection of hypo-DMGs with up-DMGs). Genes with DMRs in their promoters were significantly enriched in two biological processes, namely, developmental processes and tissue development, potentially because of the relatively lower abundance of DMRs. In contrast, genes with hypermethylation in their gene bodies showed significant enrichment in the development process, anatomical structure morphogenesis, cellular development process, and cell morphogenesis (Appendix A).

### 2.8. Validation of the Results of RNA-Sequencing Analysis by RT-qPCR

To validate the results of the RNA-sequencing (RNA-seq) analysis, seven genes were randomly selected from the DEGs to validate their expression levels in the N_IUGR and H_IUGR groups by conducting RT-qPCR. The quantitative results were normalized, and the N_IUGR group was used as the control group. Figure 9 shows the findings of the validation of the RNA-seq results. The expression patterns of all selected genes were consistent with the transcriptome sequencing results. The genes IGF2, ANXA1, SPOCK1, and ITGA8 showed significant differences in their expression levels between the H_IUGR and N_IUGR groups. Collectively, these results validated the reliability of the RNA-seq data.

## 3. Discussion

In the present study, we used high-throughput sequencing technology to investigate the genetic transcription levels and the overall methylation patterns in placenta samples from the different IUGR groups of Rongchang pigs. By constructing interaction networks and DNA methylation maps, we analyzed the dynamic regulatory molecular mechanisms and identified key candidate genes associated with IUGR. The study findings can provide valuable insights and serve as reference material to elucidate the molecular regulatory mechanisms underlying the development of Rongchang pigs.

Notably, the samples used for whole-genome bisulfite sequencing (WGBS) and RNA-seq analysis were derived from large tissues rather than single placental cells. Consequently, the measurements of DNA methylation and gene expression levels may be influenced to some extent by the presence of various cell types in placental tissues. However, according to previous studies, these potential biases are unlikely to significantly affect the main conclusions of this study [20]. Therefore, we are confident that these datasets can provide a comprehensive understanding of the potential involvement of DNA methylation in IUGR development.

The analysis of mRNA and lncRNA characteristics revealed that lncRNAs in the placental tissues of Rongchang pigs exhibited a reduced exon count, shorter ORF length, and lower expression levels as compared with mRNAs. These findings agreed with the results of previous investigations conducted on pig thyroid and endometrial tissues [21,22]. The obtained DEGs were subjected to a PPI network analysis, and three hub genes, namely, *GNGT1*, *ANXA1*, and *CDC20*, were identified. The *GNGT1* gene encodes a critical protein that is involved in the insulin resistance pathway and regulates insulin absorption through the G-protein coupling and receptor protein transduction pathways [23]. Several studies have shown the broad implication of the *GNGT1* gene in the development of diverse cancer types, thus indicating that this gene exerts a deleterious effect on the overall well-being of the organism [23,24,25]. The *ANXA1* protein regulates several cellular functions across diverse cell types, including the suppression of inflammatory responses and the facilitation of nerve repair and protection [26]. Chorioamnionitis is a primary factor in the development of IUGR. Recent research indicates that during chorioamnionitis infection, prenatal administration of multipotent adult progenitor cells enhances *ANXA1* expression in immune cells within the brain’s vascular system and barriers. This increase in ANXA1 helps protect the newborn brain from inflammation induced by infection and from additional pro-inflammatory damage in the neonatal period [27]. Neutrophils, which migrate to inflamed tissues in response to inflammatory signals, are the main source of anti-inflammatory *ANXA1*. Our findings show that *ANXA1* was significantly downregulated in H_IUGR compared with N_IUGR, implying that the low expression of *ANXA1* may lead to a lack of anti-inflammatory response, thus contributing to the occurrence of chorioamnionitis and the subsequent development of IUGR. The *CDC20* protein has a pivotal role in cell cycle regulation as it regulates the intricate process of cell mitosis [28]. CDC20 is a crucial gene influencing the formation of vascular smooth muscle. The inactivation of its associated complex can result in division disorders and cell cycle arrest in smooth muscle cells [29]. Angiogenesis is closely related to IUGR, and studies have shown that impaired vascular function can lead to fetal growth retardation and organ damage [30,31,32]. These findings suggest that CDC20 may contribute to the occurrence of IUGR by affecting angiogenesis.

Although numerous studies have elucidated the impact of gene body DNA methylation on gene expression, there is limited understanding of the precise functionality of gene body DNA methylation [33,34]. Genomic DNA methylation affects transcript elongation and splicing and intragenic promoter activity, thus playing a regulatory role in these intricate processes [35,36]. Furthermore, in certain tissues, active genes exhibit a higher degree of methylation in their gene bodies than repressed genes [37,38]. An intriguing finding of our study is the impact of methylation density on gene expression in promoter and gene body regions. Surprisingly, we observed that the methylation density in these regions does not remarkably influence gene expression. However, our investigation revealed that promoter methylation exerts a negative regulatory effect on the expression of genes involved in blood vessel development. These findings provide compelling evidence for a distinct and pivotal role of gene body methylation in IUGR development.

In this study, a comprehensive analysis revealed 495,870 DMRs. Notably, only 11,198 DMRs were found within the genes’ promoter regions, thus indicating that the majority of these DMRs are distributed outside the promoter region. This observation agreed with previous findings [39]. By integrating transcriptome and methylation data, we identified several commonly implicated pathways such as vascular development and embryogenesis. In these pathways, specific genes, including *ANXA1*, *ADM*, *NRP2*, *SHH*, and *SMAD1*, were identified as potential contributors to IUGR development. *ANXA1* belongs to the phospholipid-binding protein family and plays a crucial role in regulating cell proliferation [40]. *ADM*, originally isolated from a pheochromocytoma tissue, has been identified as a biologically active polypeptide with vasodilatory and diuretic properties [41]. *ADM* has a crucial role in regulating placental formation and maternal blood supply to the placenta. It also influences cell growth, differentiation, and angiogenesis, thereby affecting blood vessel formation [42]. ADM also can induce proliferation, migration, and adhesion in porcine trophoblast cell lines [43]. The suppression of *ADM* expression is associated with compromised embryo implantation, wherein the formation of robust blood vessels is impeded during early embryonic development. In more severe instances, this phenomenon can detrimentally affect placental blood supply and potentially culminate in fetal growth retardation. *NRP2* belongs to the neuropilin family of receptor proteins and has a critical role as a co-receptor for axon guidance factors and vascular endothelial growth factors. It also has crucial functions in cardiovascular development and the intricate processes underlying nervous system formation [44,45]. *SHH* exhibits remarkable evolutionary conservation and has a critical role in embryonic development and body tissue and organ formation. During various stages of embryonic development, *SHH* actively participates in pivotal processes. In the early embryonic development stage, *SHH* is localized in specific regions such as the nodes and notochord, where it regulates the development of the left–right and dorsal–ventral axes of the embryo. In the later embryonic development stages, *SHH* predominantly shows high expression in epithelial tissues [46]. *SHH* plays a crucial role in regulating endothelial cell growth, promoting cell migration, and stimulating the formation of new blood vessels [47]. A comprehensive analysis identified genes related to vascular development and function, such as *CDC20*, *ADM*, *NRP2*, and *SHH*, as core functional genes. This indicates that IUGR may be associated with impaired vascular development and function.

In the present study, a comprehensive methylation map of the placental tissues from Rongchang pigs was established by the WGBS method. This approach revealed distinct DNA methylation patterns between IUGR piglets and their normal counterparts, thus offering a valuable reference for future investigations on the epigenetic mechanisms underlying placental function in IUGR piglets. Moreover, a PPI network analysis performed on the differentially expressed genes across the entire transcriptome enabled us to identify the core genes in the differential regulatory network. Collectively, our study elucidates the crucial functional and regulatory role of DNA methylation in IUGR development. The methylome and transcriptome data generated could facilitate further studies to reveal the impact of DNA methylation on IUGR development and provide valuable insights for future research on animal breeding, developmental biology, and related diseases.

## 4. Materials and Methods

### 4.1. Animals

The experimental group consisted of 10 purebred Rongchang sows from the Shuanghe Farm in Rongchang, Chongqing, from the same production batch, the second parity, and with a history of IUGR. The control group consisted of 10 healthy purebred Rongchang sows from the same production batch, the second parity, and without a history of IUGR. All animals were housed under identical environmental conditions and received a standardized diet during this study. Immediately after birth, the piglets were cleaned, and their birth weights were recorded. The number of IUGR piglets was determined according to the standard, defining IUGR piglets as those whose birth weight was two standard deviations below the average birth weight of all piglets. Finally, 5 piglets with IUGR were found out of 10 sows from the experimental group, and all came from different mothers. The 5 normal piglets in the control group were randomly selected from different mothers. We designated the aforementioned five IUGR piglets as the H_IUGR group, and the five regular piglets as the N_IUGR group.

### 4.2. Tissue Collection

The placenta corresponding to each piglet was collected immediately after delivery. Tissue samples were obtained from the maternal side of the placenta at a distance approximately 2 cm from the umbilical cord insertion site; care was taken to ensure the samples were devoid of maternal decidua. To preserve the integrity of the collected placental tissues, they were rapidly frozen in liquid nitrogen. Genomic DNA and mRNA were subsequently extracted from the frozen samples and pooled for sequencing analyses.

### 4.3. Determination of DNA Methylation Degree

Genomic DNA was isolated from the retrieved placentas of piglets in each group by using the DNeasy Blood & Tissue Kit (Qiagen, Hilden, Germany). To perform WGBS analysis, total DNA was sonicated using a Covaris S220 instrument (Covaris, Woburn, MA, USA) to achieve 200 to 300 bp fragments. Subsequently, end repair and A-ligation processes were performed. Following ligation with cytosine-methylated barcodes, the DNA fragments were treated with bisulfite twice using the EZ DNA Methylation GoldTM Kit (Zymo Research, Orange, CA, USA). The libraries were then prepared in accordance with the standard DNA methylation analysis protocol provided by Illumina (Illumina, San Diego, CA, USA). RNA-seq libraries were then constructed from the same samples. Total RNA was extracted using TRIzol™ reagent (Invitrogen, Carlsbad, CA, USA), followed by rRNA depletion and DNaseI treatment (Qiagen, Hilden, Germany). For each RNA sample, a paired-end strand-specific library was constructed using the Illumina RiboZero protocol, with an insert size of approximately 350 bp. Subsequently, the WGBS and RNA-seq libraries were sequenced on the Illumina platform (Illumina, San Diego, CA, USA), and 150 bp paired-end reads were generated (Allwegene, Nanjing, China). Both the WGBS and RNA-seq libraries were obtained with five biological replicates at each developmental stage.

### 4.4. WGBS Analysis

The raw reads obtained from each WGBS library were preprocessed using Trimmomatic (version 0.39) software. Subsequently, the Sus scrofa reference genome (NCBI Sscrofa11.1) was converted to a bisulfite-treated version specifically through C-to-T and G-to-A conversion. The converted genome was then indexed using the bowtie2 alignment tool [48]. The resulting bisulfite-treated clean reads were mapped to the reference genome using Bismark software (v0.12.5) with default parameters [49]. After removing duplicate reads, the sequencing depth and coverage of methylcytosine were summarized. Our analysis focused solely on CpG methylation. The methylation level of each CpG site was determined by calculating the ratio of methylated reads (C) to the total number of methylated reads (C) and unmethylated reads (T) at the corresponding position in the reference genome. The average percent of methylation across all cytosine residues in a given genomic region was calculated as the fraction of read counts for “C” over the total read counts for both “C” and “T”. The bisulfite unconverted rate was considered the methylation level of bisulfite-treated lambda genomes.

### 4.5. Annotation of Gene and Genomic Features

The pig reference genome and gene annotation data were retrieved from the NCBI database, specifically, the S. scrofa 11.1 version. In this study, the gene body region was defined as the segment spanning from the TSS to the TES. Additionally, the promoter region was defined as the 2 kb segment located upstream of the TSS. CpGI regions were defined as segments longer than 200 bp with a GC score higher than 0.5. An observed-to-expected ratio of CpGs on repeat masked sequences greater than 0.65 indicated the presence of CpGIs. Moreover, CpGI shores and shelves were defined as 2 kb regions extending in both directions from CpGIs and CpGI shores, respectively.

### 4.6. RNA-Seq Analysis

FastQC (version 0.1.9) software was used to assess the quality of the raw data (raw reads) from each sample. To ensure data integrity, Cutadapt software (v1.9.1) was used to remove adapter sequences, eliminate bases with double-end quality values below 20, discard reads containing N bases, and filter out low-quality reads. The resulting clean read data were then aligned to the reference genome (Sscrofa11.1) obtained from the Ensembl database (http://www.ensembl.org/, accessed on 7 July 2023.) using HISAT2 (version 2.2.0) software. Subsequently, the assembled reads were processed using StringTie (version 2.1.7) software, which prioritizes the assembly of reads with successful alignment to the genome. Finally, transcriptome information for each sample was quantified.

### 4.7. Identification of DMRs and Enrichment Analysis of DMR-Related Genes

The methylation levels at individual cytosine residues were determined by calculating the ratio of methylated “C” reads to the total sequencing reads by following established protocols. Initially, as candidate sites, we selected common covered CpG sites with a sequencing depth of ≥5 between the two groups. For each cytosine residue, the methylation level was then defined as the ratio of the “C” count to the total counts of “C” and “T” in the sequencing reads. To emphasize CpG-rich regions, we considered the maximum distance of 200 bp between two adjacent CpGs and the requirement of a minimum of five CpGs in a candidate DMR, with each CpG site exhibiting a consistent methylation tendency.

### 4.8. Quantitative Real-Time PCR (RT-qPCR)

For RT-qPCR analysis, total RNA was extracted in accordance with the manufacturer’s instructions using TRIzolTM reagent (Invitrogen). The samples used were the same samples used in previous sequencing. cDNA synthesis was performed using reverse transcriptase (Bio-Rad, Hercules, CA, USA) and oligo(dT) primers. The expression levels of the DEGs were analyzed by RT-qPCR using the CFX384 multiplex real-time fluorescence quantitative PCR platform (Bio-Rad, Berkeley, CA, USA). The *ACTB* gene was used as the internal control to detect the relative expression of mRNA of *IGF2*, *ANAX1*, *CPS1*, *FOSB*, *UCP2*, *SPOCK1*, and *ITGA8* (Table 2). Each experiment was repeated thrice, and the data were presented as mean ± standard error of the mean (SEM). The relative gene expression level for each gene was determined using the 2^−ΔΔCt^ method. Appendix A provides the data on the expression levels of the genes of interest.

## Figures and Tables

**Figure 1 ijms-25-06462-f001:**
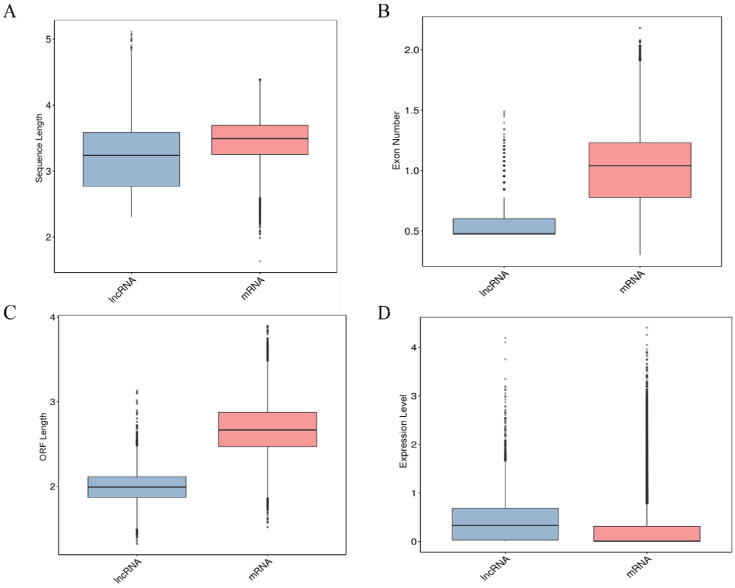
(**A**) Comparative analysis of lncRNA and mRNA length. (**B**) Comparative analysis of exon numbers between lncRNA and mRNA. (**C**) Comparative analysis of the length of the ORF of lncRNA and mRNA. (**D**) Comparison of expression levels of lncRNA and mRNA.

**Figure 2 ijms-25-06462-f002:**
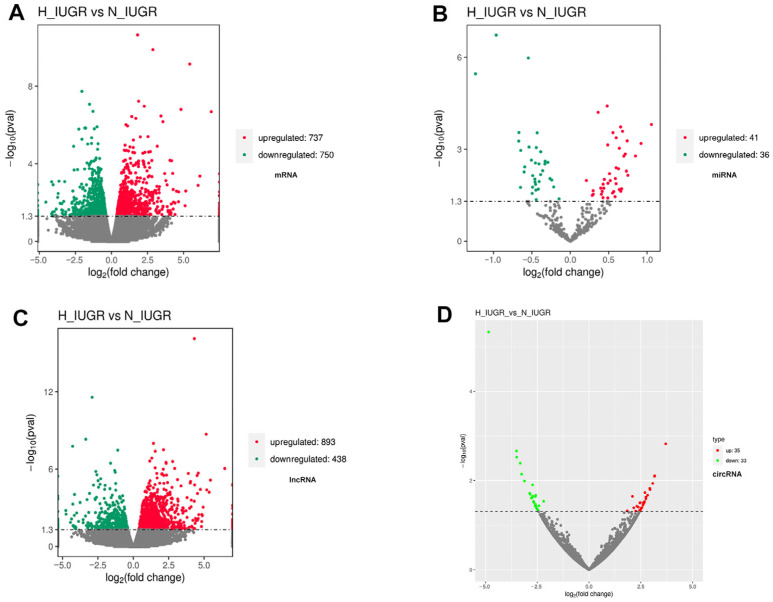
Volcano plot of differentially expressed RNAs in the H_IUGR vs. N_IUGR comparison. (**A**) mRNA, (**B**) miRNA, (**C**) lncRNA, and (**D**) circRNA.

**Figure 3 ijms-25-06462-f003:**
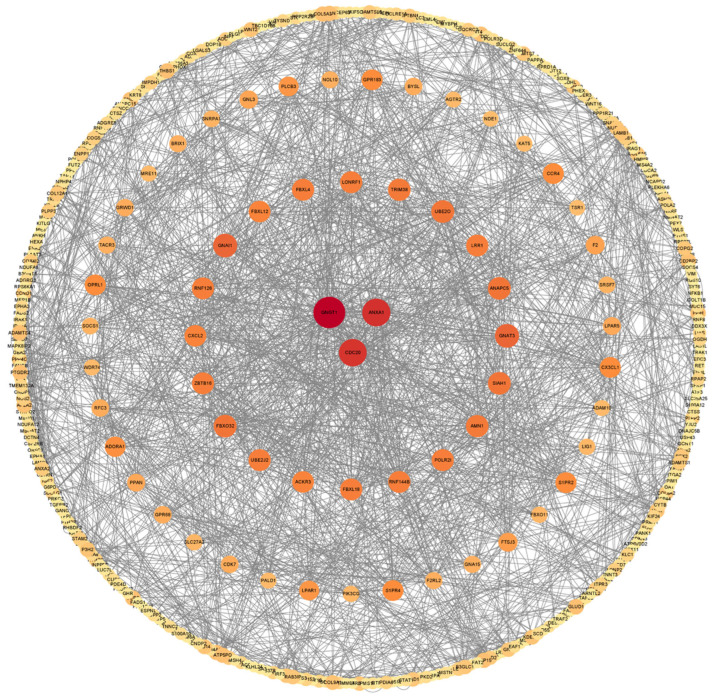
Protein–protein interaction network of target genes. Each circle node represents a gene. The circle size and color scales represent the node degree of the target genes. The darker the color and the larger the circle, the higher the core degree of the gene.

**Figure 4 ijms-25-06462-f004:**
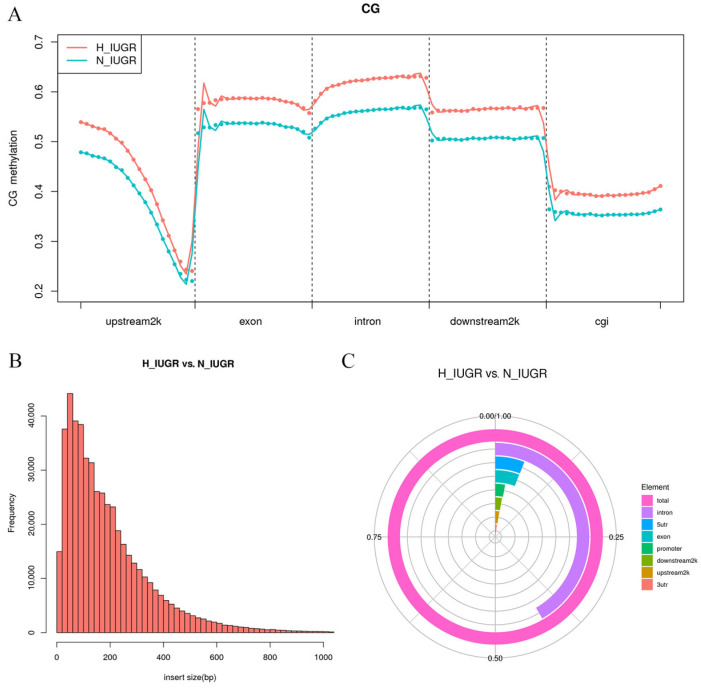
Methylation analysis results. (**A**) Distribution of methylation levels on different functional regions of the genome. The red line represents H_IUGR, and the blue line represents N_IUGR. (**B**) DMR length distribution histogram. (**C**) DMR distribution among functional elements of the genome.

**Figure 5 ijms-25-06462-f005:**
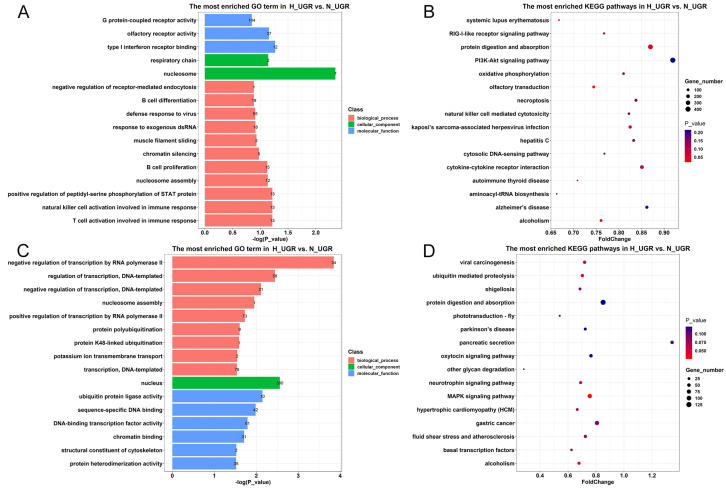
The GO and KEGG results. (**A**) GO entries mainly enriched in two placental differentially methylated genes. (**B**) KEGG pathways mainly enriched in two placental differentially methylated genes. (**C**) GO enrichment analysis of differentially methylated genes in the promoter region of two kinds of placentas. (**D**) The KEGG pathway mainly enriched in the differentially methylated genes in the promoter regions of the two placentas.

**Figure 6 ijms-25-06462-f006:**
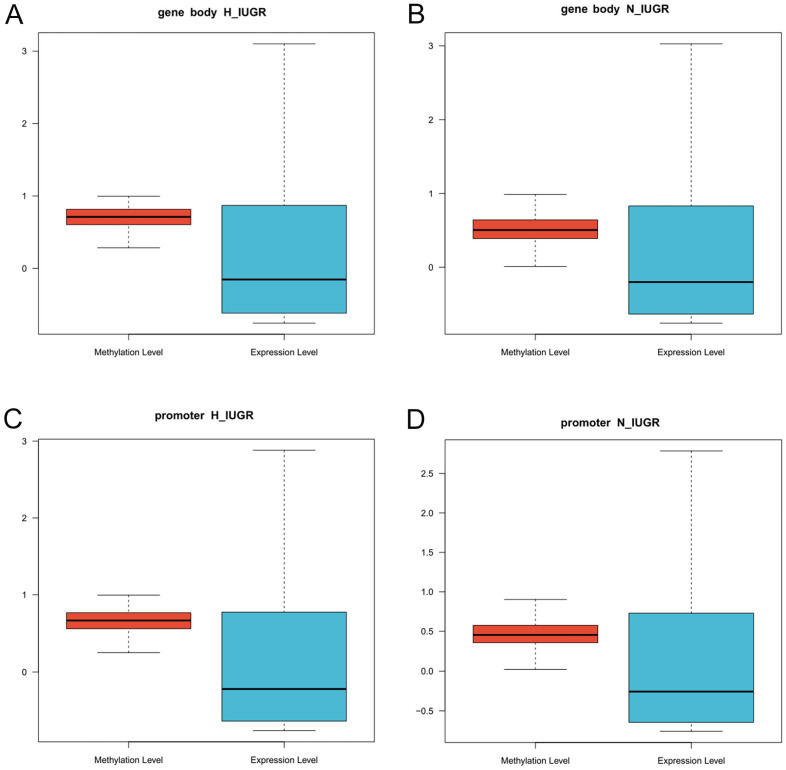
Overall correlation analysis of DNA methylation level and gene expression. (**A**) H_IUGR group gene ontology region methylation level and gene expression level. (**B**) N_IUGR group gene ontology region methylation level and gene expression level. (**C**) H_IUGR group promoter region methylation level and gene expression level. (**D**) N_IUGR group promoter region methylation level and gene expression level.

**Figure 7 ijms-25-06462-f007:**
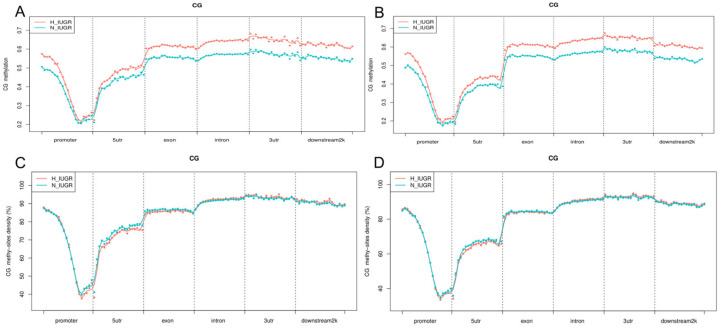
DNA methylation level and density distribution of upregulated and downregulated differentially expressed genes in different genomic elements. (**A**) Methylation levels of upregulated differentially expressed genes. (**B**) Methylation levels of downregulated differentially expressed genes. (**C**) Methylation density of upregulated differentially expressed genes. (**D**) Methylation density of downregulated differentially expressed genes.

**Figure 8 ijms-25-06462-f008:**
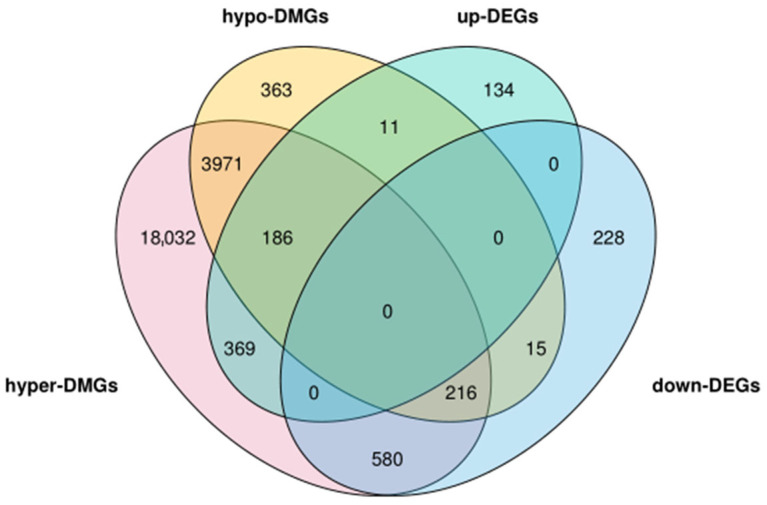
Venn diagram of differentially methylated genes and differentially expressed genes. Hyper-DMGs mean that the methylation level of the gene in the H_IUGR group was increased compared with the N_IUGR group, and the corresponding hypo-DMGs mean a low methylation level. Up-DGEs mean that compared with the N_IUGR group, the relative expression of the gene at the mRNA level in the H_IUGR group was upregulated, and vice versa for down-DGEs.

**Figure 9 ijms-25-06462-f009:**
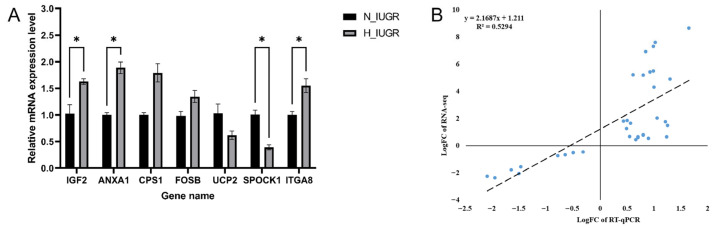
Validation of 7 DEGs between the N_IUGR and H_IUGR piglet pituitary by RT-qPCR. (**A**) The black columns indicate the N_IUGR results; the gray columns indicate the H_IUGR results. The *x*-axis represents the names of the DEGs, and the *y*-axis represents the log2 fold-change from RT-qPCR. “*” indicates *p* < 0.05. (**B**) The correlation results between the sequencing and RT-qPCR data.

**Table 1 ijms-25-06462-t001:** Statistical analysis of birth weight in two groups.

Group	Sample ID	Birth Weight (kg)	Mean of Litter Weight (kg)	Number of Litter
N_IUGR	N_IUGR 1	0.76	0.68 ± 0.09	10
	N_IUGR 2	0.80	0.68 ± 0.10	8
	N_IUGR 3	0.74	0.67 ± 0.09	11
	N_IUGR 4	0.82	0.72 ± 0.09	10
	N_IUGR 5	0.74	0.67 ± 0.10	9
H_IUGR	H_IUGR 1	0.44	0.67 ± 0.09	11
	H_IUGR 2	0.48	0.68 ± 0.10	8
	H_IUGR 3	0.46	0.68 ± 0.08	11
	H_IUGR 4	0.42	0.64 ± 0.08	9
	H_IUGR 5	0.44	0.67 ± 0.09	9

**Table 2 ijms-25-06462-t002:** Primers for RT-qPCR.

Gene	Primer Sequences (5′ to 3′)	TM (°C)	Product Size (bp)
*ACTB*	F:GATGACGATATTGCTGCGCTC R:TCGATGGGGTACTTGAGGGT	60	121
*ANXA1*	F:CTCGATTGCACTGAGGATCA R:GCTGATTCTGGCCACTTCTC	60	116
*SPOCK1*	F:GGGCTGGATGTTCAACAAGT R:CTCCAGGTTTCTTGGAAGCAG	60	105
*CPS1*	F:AAGTCCTGGGGACCTCAGTT R:AGGACAATGCCTGAGCCTA	60	113
*FOSB*	F:GTGAAGTTCAAGTCCTCGGC R:TCACAGAGCAAGAAGGGAGG	60	120
*IGF2*	F:ATCGTGGAAGAGTGCTGCTT R:CATAGCGGAAGAACTTGCCC	60	114
*UCP2*	F:GCCAACAGACGTGGTAAAGG R:TGGCATTACGAGCGACATTG	60	96
*ITGA8*	F:CTTTCACATACCTGCGGCTC R:TCAAAGGGTATCTGCCTGCA	60	154

## Data Availability

Data are available on request from the authors.

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
