# Peer review of "Maternal Hypermethylated Genes Contribute to Intrauterine Growth Retardation of Piglets in Rongchang Pigs"

_ijms, 2024, doi:10.3390/ijms25126462_

Round 1

Reviewer 1 Report

Comments and Suggestions for Authors

The authors conducted gene-wide transcriptome and methylation analyses in placental tissues from IUGR Vs normal piglets, which provides a landscape of genomic differences in Rongchang Pig. My comments are as follows.

Major questions:

1. Why authors chose to use the Rongchang Pig, is this pig breed much different, on IUGR, from other kinds of breeds, e.g. Yorkshire, Landrace and Duroc, or cross breeds?

2. On the whole, the manuscript is only descriptive results of sequencing, which is less of deeper analysis of kinds of DEGs. Information of Competitive Endogenous RNA(ceRNA) comprised  of mRNA, lncRNA, miRNAcircRNAand their relationships with DMRs/DMGs were not well summarized. Importantly, the potential regulating/determining mechanisms and/or networks in the onset of IUGR were not concluded. Enriched pathways in the manuscript were only based on the respective analysis of one kind of DEGs. So, I suggest that a better work strategy is concluding the potential regulating mechanism of IUGR first (hypothesis) using different kinds of genes, and processing further, or even preliminary, validation experiments for the hypothesis.  

3. New progress of porcine IUGR research or IUGR mechanisms in recent years (2020-2024) are lacking in the discussion section. It seems that readers could not get key points of the regulating mechanism and candidate genes with main efforts in the development of IUGR, except for some DEG descriptions based on the preliminary analysis.

Minor questions:

4. Line 25-28, descriptions of the conjoint analysis are obscure.

5. Introduction section, some reasons for IUGR, in pigs, are suggested to be described in brief. And, cases of placenta functional abnormity in the development of IUGR should be emphasized.

6. Line 92, “burn to sows”?

7. Line 98, please cite the guidelines, published articles or website.

8. Line 99, Table 1 and M&M 4.1, the animals used for sequencing were not clearly described. What is the meaning of “Sample ID”? Is it the piglets? or sows? As shown in the Table 1, there seems a total of 10 litters were used. While, there was 20 in section 4.1. Additionally, I suggest to use “Mean±SD”  for “Mean of litter weight”.

9. Line 206-207, repeated descriptions of “Ubiquitin-mediated proteolysis signaling pathway”.

10. Line 259, please use the consistent symbol for “RT-PCR” “qPCR”.

11. Line 269, meaning of “NBW”? I suggest to show fold changes of sequencing results also in the figure 9.

12. 4.2. Tissue collection, this section was not clearly described. Did the authors collected tissues for each piglets? How was the pooling strategy?

13. 4.4., I suggest authors to upload the raw data to public database, GEO or CNCB(https://ngdc.cncb.ac.cn/), and obtain the data ID number.

14. Line 433, please display “−ΔΔCt” in superscript.

15. Line 455, please keep in mind the right style of some references, such as R14-16.

16. Resolutions of figures should be improved.

Comments on the Quality of English Language

Some writings should be improved.  please see my comments for details.

Author Response

Dear Reviewers,

The authors conducted gene-wide transcriptome and methylation analyses in placental tissues from IUGR Vs normal piglets, which provides a landscape of genomic differences in Rongchang Pig. My comments are as follows.

Response: Thank you for your comments. We revised this manuscript according to referee’s detailed comments. And this revised manuscript was edited by one or more native English-speaking editors, enclosed please find the Certificate of Editing. Thank you very much for all your help and looking forward to hearing from you soon.

Major questions:

  1. Why authors chose to use the Rongchang Pig, is this pig breed much different, on IUGR, from other kinds of breeds, e.g. Yorkshire, Landrace and Duroc, or cross breeds?

ResponseThank you for your kind comments. There is an abundance of studies on Yorkshire, Landrace, and Duroc pigs, but research on Rongchang pigs is scarce. Rongchang pigs possess rich genetic diversity, and using them as a sample can yield genetic results that differ from those of more common pig breeds. This provides valuable insights for exploring how genetic factors influence IUGR from various genetic perspectives. Moreover, Rongchang pigs exhibit a high incidence of IUGR, making them particularly relevant for such studies.

  1. On the whole, the manuscript is only descriptive results of sequencing, which is less of deeper analysis of kinds of DEGs. Information of Competitive Endogenous RNA(ceRNA) comprised of mRNA, lncRNA, miRNA,circRNA,and their relationships with DMRs/DMGs were not well summarized. Importantly, the potential regulating/determining mechanisms and/or networks in the onset of IUGR were not concluded. Enriched pathways in the manuscript were only based on the respective analysis of one kind of DEGs. So, I suggest that a better work strategy is concluding the potential regulating mechanism of IUGR first (hypothesis) using different kinds of genes, and processing further, or even preliminary, validation experiments for the hypothesis.

Response: Thank you for your suggestions. We agreed that the potential regulating/determining mechanisms and/or networks in the onset of IUGR need well studied. In this research project, we aimed to identified the differently expression genes associated with IUGR, and provided the potential candidate genes for further studies. Moreover, the further regulating mechanisms and molecular function verification are intensively studied in another research project. Thus, we aimed to provide potential candidate genes contribute to IUGR development in Rongchang pigs. The further molecular modulation mechanism and function verification will show in another study.

  1. New progress of porcine IUGR research or IUGR mechanisms in recent years (2020-2024) are lacking in the discussion section. It seems that readers could not get key points of the regulating mechanism and candidate genes with main efforts in the development of IUGR, except for some DEG descriptions based on the preliminary analysis.

Response: Thank you for your comments. The new progress of porcine IUGR research in recent years was listed in the “Introduction” section.

Minor questions:

  1. Line 25-28, descriptions of the conjoint analysis are obscure.

Response: Thank you for your comments. We revised this sentence in the revised manuscript. Line 26-27

  1. Introduction section, some reasons for IUGR, in pigs, are suggested to be described in brief. And, cases of placenta functional abnormity in the development of IUGR should be emphasized.

Response: Thank you for your comments. We revised the related contents.

  1. Line 92, “burn to sows”?

Response: Thank you for your comments. We revised the problem.

  1. Line 98, please cite the guidelines, published articles or website.

Response: Thank you for your comments. We added the reference in the revised manuscript.

  1. Line 99, Table 1 and M&M 4.1, the animals used for sequencing were not clearly described. What is the meaning of “Sample ID”? Is it the piglets? or sows? As shown in the Table 1, there seems a total of 10 litters were used. While, there was 20 in section 4.1. Additionally, I suggest to use “Mean±SD” for “Mean of litter weight”.

Response: Thank you for your comments. We revised this manuscript. The experimental group consisted of 10 purebred Rongchang sows from the Shuanghe Farm in Rongchang, Chongqing, from the same production batch, the second parity, and with a history of IUGR. The control group consisted of 10 healthy purebred Rongchang sows from the same production batch, the second parity, and without a history of IUGR. All animals were housed under identical environmental conditions and re-ceived a standardized diet during the study. Immediately after birth, the piglets were cleaned and their birth weights were recorded. The number of IUGR piglets was deter-mined according to the standard, defining IUGR piglets as those whose birth weight is two standard deviations below the average birth weight of all piglets. Finally, 5 piglets with IUGR were found out of 10 sows form experimental group, and came from different mothers. The 5 normal piglets in the control group were randomly selected different mothers. We had designated the aforementioned five IUGR piglets as the H_IUGR group, and the five regular piglets as the N_IUGR group.

  1. Line 206-207, repeated descriptions of “Ubiquitin-mediated proteolysis signaling pathway”.

Response: Thank you for your comments. We deleted the descriptions of “Ubiquitin-mediated proteolysis signaling pathway”.

  1. Line 259, please use the consistent symbol for “RT-PCR” “qPCR”.

Response: Thank you for your reminder. It has been revised.

  1. Line 269, meaning of “NBW”? I suggest to show fold changes of sequencing results also in the figure 9.

Response: Thank you for your reminder. It has been revised. The fold changes of sequencing results were descript in the previous content.

  1. 4.2. Tissue collection, this section was not clearly described. Did the authors collected tissues for each piglets? How was the pooling strategy?

Response: Thank you for your comments. This contents were listed in the “Materials and Methods” section. Please see “4.1 Animals”.

  1. 4.4., I suggest authors to upload the raw data to public database, GEO or CNCB(https://ngdc.cncb.ac.cn/), and obtain the data ID number.

Response: Thank you for your suggestions. The raw data will be uploaded to public database in future. At present, the data are available from the corresponding authors upon reasonable request.

  1. Line 433, please display “−ΔΔCt” in superscript.

Response: Thank you for your reminder. It has been revised.

  1. Line 455, please keep in mind the right style of some references, such as R14-16.

Response: Thank you for your reminder. It is not reference. The R14-16 is the experimental procedure.

  1. Resolutions of figures should be improved.

Response: Thank you for your comments. The figures were revised in the revised manuscript.

Pingxian Wu

Sincerely.

Reviewer 2 Report

Comments and Suggestions for Authors

The authors have taken a good initiative to conduct an interesting study. Assessing the molecular mechanism associated with intrauterine growth retardation would aid in resolving uncertain concepts.

There are a few concerns and suggestions on some aspect in this manuscript which have been listed below.

Major comments:

1.     The authors have taken a good effort to correlate the DNA methylation and transcriptomics data. However it would have been more interesting if they could enlist a few potential genes showing characteristic methylation/expression profiles

2.     The discussion may be improved. The authors haven’t discussed the results obtained from transcriptomics and metagenomics (only the correlated analysis was discussed). Likewise, drawing a more clear and definite conclusion on the probable mechanism playing a vital role in intrauterine growth retardation would be appreciated

3.     Materials and methods: The authors have stated that “Genomic DNA and mRNA were subsequently 362 extracted from the frozen samples and pooled for sequencing analyses.” And later on have also mentioned “Both 378 the WGBS and RNA-seq libraries were obtained with three biological replicates at each 379 developmental stage.” This is confusing, kindly state clearly if the NGS performed were on pooled samples or distinct biological samples.

4.     Likewise in table 1 (line 99), the statistical analysis depicts 10 animals while in the materials and methods (line 354) the total number of animals mentioned was 20. Kindly address this discrepancy

5.     Also clearly state the total number of samples/animals taken for NGS

6.     Line 425 (4.8. Quantitative real-time PCR (qPCR)): were the samples (animal ID and tissue type) selected for validation the same as those sequences?

7.     Line 425 (4.8. Quantitative real-time PCR (qPCR)): Kindly mention the genes considered for validation. Also in supplementary table 2 (primers for RT-qPCR), kindly indicate the primer sequence for the reference primer (ACTB)

8.     Did the authors submit the whole transcriptome and bisulfite sequencing data to NCBI, doing so is often suggested

Minor comments:

1.     Kindly indicate all gene names in italics

2.     Line 80: seems like there was a typing error when using the term “gene-wide transcriptome” which would be indented to be mentioned as either “genome-wide transcriptomics” or “whole transcriptome”, etc.

3.     Line 98: kindly indicate the reference for the stated guideline

4.     Line 233: Kindly indicate the reference for the ‘previous research’ mentioned in the sentence

5.     Line 274-279: kindly indicate the reference to the stated previous study

Author Response

Dear Reviewers,

The authors have taken a good initiative to conduct an interesting study. Assessing the molecular mechanism associated with intrauterine growth retardation would aid in resolving uncertain concepts.

There are a few concerns and suggestions on some aspect in this manuscript which have been listed below.

Response: Thank you for your comments. We revised this manuscript according to referee’s detailed comments. And this revised manuscript was edited by one or more native English-speaking editors, enclosed please find the Certificate of Editing. Thank you very much for all your help and looking forward to hearing from you soon.

In addition to these general changes, I have some specific comments:

Major comments:

  1. The authors have taken a good effort to correlate the DNA methylation and transcriptomics data. However it would have been more interesting if they could enlist a few potential genes showing characteristic methylation/expression profiles

Response: Thank you for your suggestions. In this research project, we aimed to identified the differently expression genes associated with IUGR, and provided the potential candidate genes for further studies. Moreover, the further regulating mechanisms and molecular function verification are intensively studied in another research project. Thus, we aimed to provide potential candidate genes contribute to IUGR development in Rongchang pigs. The further molecular modulation mechanism and function verification will show in another study.

  1. The discussion may be improved. The authors haven’t discussed the results obtained from transcriptomics and metagenomics (only the correlated analysis was discussed). Likewise, drawing a more clear and definite conclusion on the probable mechanism playing a vital role in intrauterine growth retardation would be appreciated

Response: Thank you for your comments. We revised this discussion in the revised manuscript.

  1. Materials and methods: The authors have stated that “Genomic DNA and mRNA were subsequently 362 extracted from the frozen samples and pooled for sequencing analyses.” And later on have also mentioned “Both 378 the WGBS and RNA-seq libraries were obtained with three biological replicates at each 379 developmental stage.” This is confusing, kindly state clearly if the NGS performed were on pooled samples or distinct biological samples.

Response: Thank you for your comments. It has been revised in line470.

  1. Likewise in table 1 (line 99), the statistical analysis depicts 10 animals while in the materials and methods (line 354) the total number of animals mentioned was 20. Kindly address this discrepancy

Response: Thank you for your comments. It has been revised.

  1. Also clearly state the total number of samples/animals taken for NGS

Response: Thank you for your comments. This contents were listed in the “Materials and Methods” section.

  1. Line 425 (4.8. Quantitative real-time PCR (qPCR)): were the samples (animal ID and tissue type) selected for validation the same as those sequences?

Response: Thank you for your comments. It has been revised.

  1. Line 425 (4.8. Quantitative real-time PCR (qPCR)): Kindly mention the genes considered for validation. Also in supplementary table 2 (primers for RT-qPCR), kindly indicate the primer sequence for the reference primer (ACTB)

Response: Thank you for your comments. It has been revised.

  1. Did the authors submit the whole transcriptome and bisulfite sequencing data to NCBI, doing so is often suggested

Response: Thank you for your suggestions. The raw data will be uploaded to public database in future. At present, the data are available from the corresponding authors upon reasonable request.

Minor comments:

  1. Kindly indicate all gene names in italics

Response: Thank you for your comments. It has been revised.

  1. Line 80: seems like there was a typing error when using the term “gene-wide transcriptome” which would be indented to be mentioned as either “genome-wide transcriptomics” or “whole transcriptome”, etc.

Response: Thank you for your comments. It has been revised.

  1. Line 98: kindly indicate the reference for the stated guideline

Response: Thank you for your comments. We added the reference.

  1. Line 233: Kindly indicate the reference for the ‘previous research’ mentioned in the sentence

Response: Thank you for your comments. We deleted this sentence.

  1. Line 274-279: kindly indicate the reference to the stated previous study

Response: Thank you for your comments. We added the reference line 353.

Pingxian Wu

Sincerely.

Reviewer 3 Report

Comments and Suggestions for Authors

In this manuscript, Wang et al. aim to understand the molecular relationships involved in IUGR in Rongchang pigs. The authors employed next-generation sequencing followed by a series of bioinformatic analyses to elucidate the molecular causes of IUGR. I appreciate the effort put into the sample processing and data analysis. However, although the results show some significance, the overall writing needs improvement for better understanding of the study's concepts and significance. Each results section needs rewriting to clarify why the analyses were conducted to understand IUGR. The lack of clarity in all results sections diminishes the perceived significance of the research. The writing flow is also poor.

In addition to these general changes, I have some specific comments:

Results Section 2.2: Please explain why this section is important in this article. I didn’t see any significant insights from these comparative analyses. Additionally, the y-axis of the graph is not properly labeled.

    The graphs in Figure 2 are not uniform despite using the same analysis methods, including fonts and style.

    There are typos in line 206 ("Ubiquitin-mediated proteolysis signaling pathway" is repeated).

    The labeling in Figure 5 is unclear.

    I didn’t understand the concept of the analysis in Figure 6.

    The purpose of the Venn diagram analysis needs to be clarified.

Author Response

Dear Editor and Reviewers,

Thank you very much for your comments. We have revised the manuscript according to your kind advices. And, we carefully improve the quality of English. This revised manuscript was edited by one or more native English-speaking editors, enclosed please find the Certificate of Editing. Thank you very much for your help and looking forward to hearing from you soon.

In this manuscript, Wang et al. aim to understand the molecular relationships involved in IUGR in Rongchang pigs. The authors employed next-generation sequencing followed by a series of bioinformatic analyses to elucidate the molecular causes of IUGR. I appreciate the effort put into the sample processing and data analysis. However, although the results show some significance, the overall writing needs improvement for better understanding of the study's concepts and significance. Each results section needs rewriting to clarify why the analyses were conducted to understand IUGR. The lack of clarity in all results sections diminishes the perceived significance of the research. The writing flow is also poor.

Response: Thank you for your comments. We revised this manuscript according to referee’s detailed comments. And this revised manuscript was edited by one or more native English-speaking editors, enclosed please find the Certificate of Editing. Thank you very much for all your help and looking forward to hearing from you soon.

In addition to these general changes, I have some specific comments:

Moreover, in this research project, we aimed to identified the differently expression genes associated with IUGR, and provided the potential candidate genes for further studies. Moreover, the further regulating mechanisms and molecular function verification are intensively studied in another research project. Thus, we aimed to provide potential candidate genes contribute to IUGR development in Rongchang pigs. The further molecular modulation mechanism and function verification will show in another study.

Results Section 2.2: Please explain why this section is important in this article. I didn’t see any significant insights from these comparative analyses. Additionally, the y-axis of the graph is not properly labeled.

Response: Thank you for your comments. Due to the log-transformed, The y-axis of the graph is properly labeled. In order to compare the Transcription characteristics between H_IUGR and N_IUGR group, the structure and expression level of lncRNA and mRNA were compared, and we analyzed the comparison of transcript length, exon number, ORF length and expression level between lncRNA and mRNA. On the one hand, the molecular differences between lncRNA and mRNA were obtained, and on the other hand, whether the predicted lncRNA molecules were consistent with the general characteristics was verified.

The graphs in Figure 2 are not uniform despite using the same analysis methods, including fonts and style.

Response: Thank you for your comments. It has been revised.

    There are typos in line 206 ("Ubiquitin-mediated proteolysis signaling pathway" is repeated).

Response: Thank you for your comments. It has been revised.

    The labeling in Figure 5 is unclear.

Response: Thank you for your comments. It has been revised.

    I didn’t understand the concept of the analysis in Figure 6.

Response: Thank you for your comments. The correlation coefficients between the DNA methylation and differentially expressed genes were used to evaluate the relationship between DNA methylation and differentially expressed. Because DNA promoter methylation will inhibit gene expression, and the gene ontology methylation will promote gene expression.

    The purpose of the Venn diagram analysis needs to be clarified.

Response: Thank you for your comments. It has been clarified in the revised manuscript.

Pingxian Wu

Sincerely.

Round 2

Reviewer 1 Report

Comments and Suggestions for Authors

Thanks for your revised manuscript. It has been much improved than the 1st version. 3 more comments for the revised paper.

1) Still, it seems the authors did not figure out my 2nd question, and, this is the key point in this study. In fact, the authors can do it with improved/deeper data analysis instead of new experimental works.

2) The resolution of Fig.3 is still too low.

3) Please add the correlation results between the Sequencing data and RT-qPCR data as "B" in the Fig.9.    

Author Response

Dear Editor and Reviewers,

Thank you very much for your comments. We have revised the manuscript according to your kind advice. Thank you very much for your help and looking forward to hearing from you soon.

Thanks for your revised manuscript. It has been much improved than the 1st version. 3 more comments for the revised paper.

1) Still, it seems the authors did not figure out my 2nd question, and, this is the key point in this study. In fact, the authors can do it with improved/deeper data analysis instead of new experimental works.

ResponseThank you for your suggestions. We agreed with your advice, but the further regulating mechanisms and molecular function verification are studying in another research project. Thus, we mainly provide potential candidate genes contribute to IUGR development in Rongchang pigs.

2) The resolution of Fig.3 is still too low.

Response: Thank you for your comments. We provided the high-definition image with 300dpi in the revised manuscript.

3) Please add the correlation results between the Sequencing data and RT-qPCR data as "B" in the Fig.9.  

Response: Thank you for your comments. We add the correlation results between the Sequencing data and RT-qPCR data as "B"  in the Fig.9.  

Sincerely,

Pingxian Wu

Reviewer 2 Report

Comments and Suggestions for Authors

The authors have taken a good effort on revising the manuscript and clearing the confusion. 

I only have one minor suggestion:

Line 414-415: Kindly avoid using the word ‘pooled’ when the samples were sequenced separately without pooling.

Author Response

Dear Editor and Reviewers,

Thank you very much for your comments. We have revised the manuscript according to your kind advice. Thank you very much for your help and looking forward to hearing from you soon.

The authors have taken a good effort on revising the manuscript and clearing the confusion. 

I only have one minor suggestion:

Line 414-415: Kindly avoid using the word ‘pooled’ when the samples were sequenced separately without pooling.

Response: Thank you for your comments. We revised this problem.

Sincerely,

Pingxian Wu

Reviewer 3 Report

Comments and Suggestions for Authors

The revised version of this manuscript has improved in some areas; however, my main concern regarding Figures 2.2 and 6 has not been clearly addressed. In Figure 2.2, the comparison is made between lncRNA and mRNA. It is well established that lncRNA is completely different from mRNA. In my view, this analysis is an apple-to-orange comparison. I would expect these metrics to compare H_IUGR and N_IUGR instead. The same expectation applies to Figure 6.

Author Response

Dear Editor and Reviewers,

Thank you very much for your comments. We have revised the manuscript according to your kind advices. Thank you very much for your help and looking forward to hearing from you soon.

The revised version of this manuscript has improved in some areas; however, my main concern regarding Figures 2.2 and 6 has not been clearly addressed. In Figure 2.2, the comparison is made between lncRNA and mRNA. It is well established that lncRNA is completely different from mRNA. In my view, this analysis is an apple-to-orange comparison. I would expect these metrics to compare H_IUGR and N_IUGR instead. The same expectation applies to Figure 6.

Response: Thank you for your comments. The genetic mechanisms underlying IUGR development remain unclear, it is important to clarify genetic differences. And, this is the first study to identify candidate genes associated with IUGR in Rongchang pigs. The Figure 2.2 would provide transcription characteristics and genetic difference between H_IUGR and N_IUGR group. Moreover, the Figure 6 reveled that the gene with high methylation levels were associated with lower gene expression levels, and implied DNA methylation of gene-body and promoter regions result in repression of gene expression in IUGR pigs. The obtained results provide new insight into genetic architecture in IUGR pigs. However, these results need further validate these identified genes.

Sincerely,

Pingxian Wu

Round 3

Reviewer 1 Report

Comments and Suggestions for Authors

Thanks, I don't have any questions.

Reviewer 3 Report

Comments and Suggestions for Authors

Revised comments acceptable for publication